# METRIC TRANSFORM: EXPLORING BEYOND AFFINE TRANSFORM FOR NEURAL NETWORKS

## ABSTRACT

Artificial Neural Networks(ANN) of varying architectures are generally paired with linear transformation at the core. However, we find dot product neurons with global influence less interpretable as compared to a more local influence of euclidean distance (as used in RBF). In this work, we explore the generalization of dot product neurons to lp-norm, metrics, and beyond. We find such metrics as transform performs similarly to affine transform when used in MLP or CNN. Furthermore, we use distance/similarity measuring neurons to interpret and explain input data, overfitting and Residual MLP. *We share our code in <github>*.

## 1 INTRODUCTION AND RELATED WORKS

Neural Networks are used end-to-end and generally as black-box function approximators. This is partly due to the vast number of parameters, the underlying function used, and the high dimension of input and hidden neurons. The backbone of Deep Networks including MLP, CNN Krizhevsky et al. (2012), Transformers Vaswani et al. (2017), and MLP-Mixers Tolstikhin et al. (2021) has been a linear transform of form $\mathbf{Y} = \mathbf{XW} + \mathbf{b}$ (or per neuron: $y_i = (\mathbf{x}.\mathbf{w} + b)$.

There have been explorations of other operations such as $l^1$-norm Chen et al. (2020) and $l^2$-norm Li et al. (2022) for transformations in Neural Networks much guided by the computational efficiency. However, matrix multiplication has won a hardware-lotteryHooker (2021), is highly optimized and other transformations are not explored much. In this work, we explore a form of norms and generalized measure of distance as a neuron, which we call metrics - of the form $y_i = f_{metric}(\mathbf{x}\text{-}\mathbf{w})$ where $f_{metrics}$ can be $l^p$-norm or any generalization. We also use the $f_{metrics}$ for measuring similarity as a negative or inverse distance.

**Interpretation of Neurons in 2D:** Although simple, the dot product is difficult to comprehend. It represents a planar neuron rather than a local neuron as shown in Figure( 1 *left*). Local neurons are generally found on Radial Basis Function (RBF), however, the radial function is euclidean. Furthermore, such transforms are not used in ANN.

This motivates us to explore the area of metrics that give a sense of distance between input and weights in neural networks.

## 2 EXPLORATION

We first explore $l^p$-norm. However, we find it difficult to optimize without normalization like layer-norm Ba et al. (2016) or Softmax. We add layer-norm after distance and find that $l^{0.5}$, $l^1$, $l^2$, $l^{20}$ norms along with *stereographic projection + linear* works as a transform in ANN. Experiments on 2D classification, regression, and Fashion-MNIST dataset show that these methods work better or worse than linear transforms. In these datasets, we also test with learnable metrics using convex Amos et al. (2017), invex Sapkota & Bhattarai (2021); Nesterov et al. (2022), and ordinary Neural Networks. The accuracy on Fashion-MNIST using various metrics of hidden units 20 is: $l^{0.5} = 75.54\%$, $l^1 = 80.08\%$, $l^2 = 83.82\%$, $l^{20} = 85.17\%$, stereo= 85.86%, linear= 86.27%, convex= 81.03%, invex= 88.85% and ordinary= 85.89%.

We also extend the transform for CNN (ResNet-20) and find that $l^2$-norm = 92.64%, *spectreographic projection*= 90.51% work on the CIFAR-10 dataset. The accuracy for linear CNN is 92.96%.

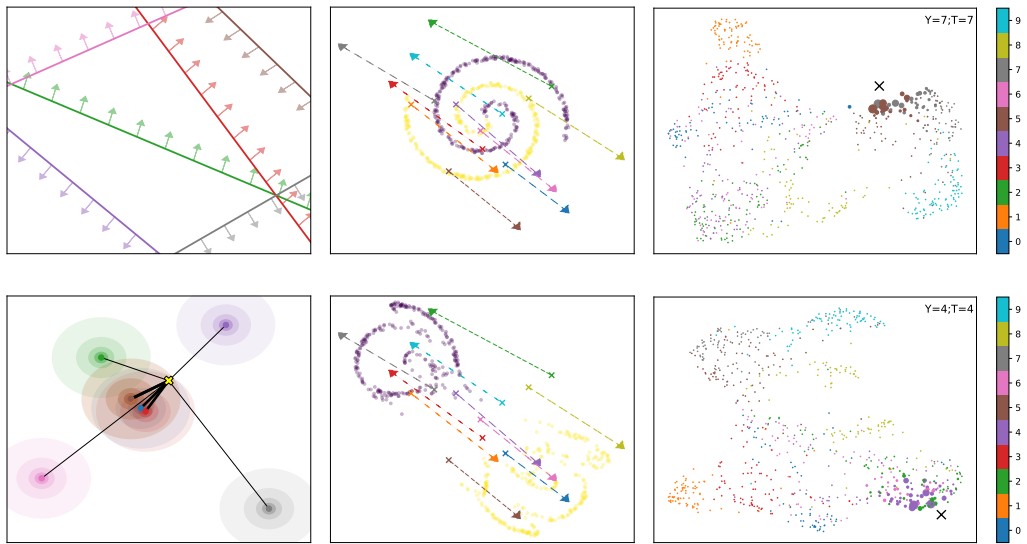

Figure 1: **Left:** Neuron Interpretation *(TOP)* 2D ReLU neuron at $\boldsymbol{w}.\boldsymbol{x} \geq 0$ showing the region of neuron firing. *(BOT)* 2D radial neuron at $f_{metric}(\boldsymbol{x} - \boldsymbol{w}) \geq \epsilon$ showing region of neuron firing. ($\times$) showing a data point and its similarity to centers. **Middle:** Local Residual Layer interpretation; visualization of the dataset ($\cdot$) along with centers ($\times$); Residual MLP moves the centers as shown by the arrow. *(TOP)* $\mathbf{x}$-space *(BOT)* $\mathbf{x} + f_{res}(\mathbf{x}) = \mathbf{y}$ space. **Right:** Data Interpretation. Points represent the UMAP of weights/center and the size represents similarity to the test sample ($\times$). Y and T represent the output of the model and target respectively. Class colors are the prediction of the given weights. *(TOP)* using dot-product weights *(BOT)* using l2-norm centers. Zoom in for details.

**Local Residual MLP:** We also interpret the residual network of form $\mathbf{y} = \mathbf{x} + \mathbf{W_1} \cdot f_{metric}(\mathbf{x} - \mathbf{W})$ . We find that the peak activation of $\approx \mathbf{1}$ is produced by similarity $f_{metric}$ neuron when the center and data are similar. The vector $\mathbf{w_1}$ for each neuron represents the shift of $\mathbf{x}$-space after residual as shown in Figure( 1 *middle*).

**Overfitting 2-layer MLP:** We are easily able to overfit a metric-transform based 2-layer neural network by using M neurons for M data points. However, this is not feasible for a large dataset. We experiment with center initialization in $l^2 - norm$ ANN to some random training samples to find that we can gain significant accuracy without even training. In 2-layer FMNIST, the number of *neuron* and *accuracy* pair is given as $\{ 10 : 37.08\%, 50 : 58.11\%, 200 : 67.64\%, 1000 : 73.16\%, 5000 : 73.98\%\}$. Similarly for MNIST dataset: $\{ 10 : 38.92\%, 50 : 60.02\%, 200 : 75.01\%, 1000 : 84.47\%, 5000 : 88.82\%\}$.

**Interpretation of High Dimensional Data:** UMAPMcInnes et al. (2018) has been used widely to visualize high dimensional dataset. However, it does not contain all the information to reconstruct inputs from embeddings. To this end, we plot UMAP embedding of centroids and add extra dimension to have magnitude of activations as shown in Figure( 1 *right*). Here, we use negative exponential to get local similarity measure. The information of activation gives unique values for each data points, and can even be invertible (see Appendix C). The mapping of centers in low dimension along with magnitude of activation allows us to interpret high-dimensional input space in embedding space.

## 3 DISCUSSION AND CONCLUSION

We find that various metrics also work and also provide transformation easily understandable in space. The transformation is done on point space rather than on vector space (as done by linear transform) which is rather difficult to understand. Our goal is not to replace linear transform but to find ways to overcome the limitations of linear neurons that have global influence and move towards more local activating neurons. Our exploration shows that concept of locality, similarity and distance is important to interpret ANNs.

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

## A  EXPERIMENT: FASHION-MNIST METRICS

We experiment with varying hidden units in FMNIST 2 layer MLP with first layer a metric and second layer linear. The detailed accuracy is listed below for various hidden units.

For hidden units 10 is: $l^{0.5} = 72.08\%$, $l^1 = 77.91\%$, $l^2 = 82.35\%$, $l^{20} = 83.98\%$, stereo$= 84.73\%$, linear$= 84.89\%$, convex$= 78.99\%$, invex$= 88.41\%$ and ordinary$= 86.11\%$.

For hidden units 100 is: $l^{0.5} = 78.27\%$, $l^1 = 82.07\%$, $l^2 = 86.41\%$, $l^{20} = 87.02\%$, stereo$= 87.99\%$, linear$= 88.67\%$, convex$= 78.07\%$, invex$= 87.85\%$ and ordinary$= 87.25\%$.

Here, the learnable convex, invex and ordinary metric functions have 2 layer architecture with 500 hidden units.

## B  HIGH DIMENSIONAL DATA VISUALIZATION

We add extra set of visualizations of high dimensional data: F-MNIST in this section.

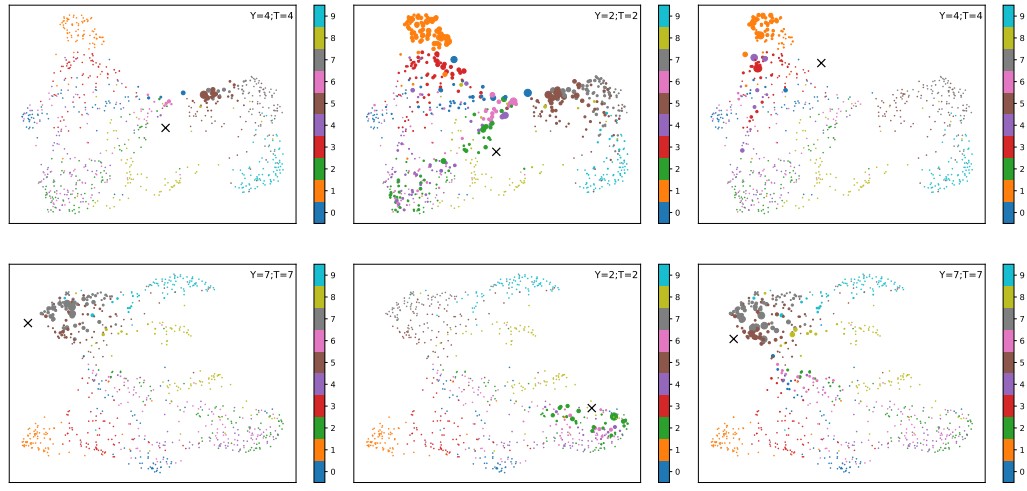

Figure 2: Data Interpretation. Points represent the UMAP of weights/center and the size represents similarity to the test sample ($\times$). Y and T represent the output of the model and target respectively. Class colors are the prediction of the given weights. *(TOP)* using dot-product weights *(BOT)* using l2-norm centers. Zoom in for details.

## C  INVERTING EUCLIDEAN DISTANCE

Inverting a measure of distance from centers is inspired by the Global Positioning System (GPS) Bancroft (1985); Norrdine (2012). These system use distance from 4 known centers to locate a point in 3D space. We find euclidean distance can be inverted in any dimensions. If the dimension of space/point is $N$, then we need $N + 1$ distances from known points to be able to reconstruct it. We also develop an algorithm to invert N-Dimensional Euclidean Distance Transform, shared with the code.

*We also hypothesize that: different metric functions, including general convex distance, can be inverted or are bijective; the invex function are injective; and ordinary function are many-to-many functions. However, we lack the proof for this hypothesis.*

