# OpenReview forum: "Metric Transform: Exploring beyond Affine Transform for Neural Networks"
_ICLR.cc/2023/TinyPapers — Submitted to Tiny Papers @ ICLR 2023_

### Official Review · Reviewer_Ja6Z · 2023-03-19

**Confidence:** 4

**Summary Of Contributions:**

This paper proposes to replace the affine transformations used in most neural networks with some more general transformations.

**Rating:**

Needs Clarification (NC): a submission which does not meet the reviewing criteria and needs clarification for its described problem or solution

**Strengths And Weaknesses:**

I summarize the strengths and weaknesses of this submission with respect to the each of reviewing criteria below.

- **Clarity**: Though the idea of incorporating more complex single neurons into a deep network is clear, the motivation this work gives for doing so is not clear to me. The authors write "This motivates us to explore the area of metrics that give a sense of distance between input and weights in neural networks." I don't see how this motivates moving beyond inner products, as they of course give a sense of distance between inputs and weights - they compute the un-normalized cosine distance, which is a sort of template-matching. The authors don't offer any reasoning behind their assertion that the dot product is "difficult to comprehend."

- **Correctness**: As mentioned above, I don't think the author's basic premise that dot products are "difficult to comprehend" is adequately justified. I also have some questions regarding the experiments presented, which I detail under Suggested Changes.

- **Reproducibility**: The authors promise to release code. That would go a long way towards improving reproducibility, as the current manuscript does not provide adequate description of the architectures listed.

- **Follows basic requirements**: The paper appears to conform to the length and anonymity requirements.

**Suggested Changes:**

1. The authors assert that "although simple, the dot product is difficult to comprehend[; it] represents a planar neuron rather than a local neuron." I find this neither clear nor convincingly argued. What about the inner product, or the cosine distance, is so challenging conceptually? To make this point clear and convincing, the authors must provide more justification for why they think RBF neurons are easier to grok.

2. In the computational neuroscience community, single-neuron complexity beyond the point neuron has previously been motivated from the perspective of nonlinear dendritic computation. To help contextualize their work, the authors might consider mentioning this line of work, as reviewed, for instance, in Chavlis & Poirazi, "Drawing inspiration from biological dendrites to empower artificial neural networks," *Current Opinion in Neurobiology* 2021.

3. The role of activation function is not described clearly. Yes, most neural networks are based on affine transformations, but those affine transformations are in turn composed with pointwise nonlinearities. What activation functions did the authors use in their experiments in Section 2? Without this, it is difficult to contextualize the complexity added by $f_{\textrm{metric}}$.

4. What restrictions do you impose on the function $f_{\textrm{metric}}$? Do you require the output $f_{\textrm{metric}}(\mathbf{x}-\mathbf{w})$ to be an honest distance metric? Moreover, as written, the family of transformations considered here do not appear to include affine transformations as a special case.

5. The UMAP visualizations should be explained in more detail. What is it that you aim to show? I was puzzled by the statement that "we plot UMAP embedding of centroids and add extra dimension to have magnitude of activations." It does not seem that only class centroids are being embedded. Also, is it the activation of a single neuron that determines the size of each point? If so, how is that neuron chosen?

---

### Official Review · Reviewer_6eDZ · 2023-03-23

**Confidence:** 3

**Summary Of Contributions:**

The paper proposes an alternative layer for neural networks, by using metrics (such as L2 norm) to calculate neuron outputs instead of the classic linear operation (x*W). The goal is to localize neuron operations, in order to improve interpretability.

**Rating:**

Needs Clarification (NC): a submission which does not meet the reviewing criteria and needs clarification for its described problem or solution

**Strengths And Weaknesses:**

### Strengths
1. High number of experiments: the authors provide results for many experiments.
1. Nice idea to improve interpretability.
### Weaknesses
1. Many sentences are hard to follow and understand.
e.g last line of introduction (before "interpretation of Neurons in 2D"): "We also use the f_metrics for measuring similarity as a negative or inverse distance", this sentence is never explained. Some sentences are grammatically incorrect such as "[...] various metrics also work and also provide transformation easily [...]".
1. No example is given of the output of their "new neuron", and their model is never presented in detail. The authors claim that a repository with code is available, but without it (redacted for double blind) it's very hard to understand the operations carried out by the model.
1. In the experiments, they present a "linear" layer, which I assume to be the default layer calculating $y = xW + b$, but then also present a "ordinary Neural Netwoks" layer, which I also interpret as being the same linear layer (since no explanation is given) but it has different results compared to the "linear" one. This creates confusion since now I don't know what "linear" means in the context of this paper.
1. The paper claims that their results show that the "concept of locality, similarity and distance is
important to interpret ANNs", however this is never explored and/or explained with practical examples (fig. 1 is not clear enough).

**Suggested Changes:**

1. Provide a scheme of the model and/or layers.
1. Provide a scheme of the data-flow inside the network, with input/output shapes for each "novel" layer and an explanation of what the output is.
1. The previous suggestions should replace some sections of the paper that are poorly written/explained, or that do not contribute to better explain the method (e.g. fig.1)
1. Tabularize the results
1. Further analyze the interpretability of your approach.

---

### Meta-Review · Area_Chair_5dqR · 2023-04-07

**Recommendation:** Invite to revise
**Confidence:** 4

**Metareview:**

The paper proposes a novel layer for neural networks that calculates neuron outputs using metrics instead of the classic linear operation. Both reviewers agree that the idea has potential for improving interpretability, but the paper lacks clarity and organization. Reviewer 6eDZ notes that the paper is poorly written and lacks details, while Reviewer Ja6Z questions the justification for moving beyond inner products and requests more explanation for the experiments presented. Overall, the paper has potential but requires significant revisions.


**Summary:**

The paper proposes a novel layer for neural networks that calculates neuron outputs using metrics instead of the classic linear operation, with the goal of improving interpretability.

**Comments And Feedback To The Authors:**

- Provide a detailed explanation of the proposed model and experiments, including a scheme of the data-flow inside the network with input/output shapes for each "novel" layer and an explanation of what the output is.
- Justify the motivation for moving beyond inner products and provide more details about the role of activation functions used in the experiments.
- Consider mentioning the previous work on incorporating more complex single neurons into deep networks motivated from the perspective of nonlinear dendritic computation to contextualize the proposed work.


**Reason For Not Giving A Higher Recommendation:**

The submission is rated as Needs Clarification by both reviewers, indicating that it does not meet the reviewing criteria and requires significant revisions. Both reviewers suggest that the paper lacks clarity and organization and that important details are missing, such as a detailed explanation of the proposed model and experiments.

**Reason For Not Giving A Lower Recommendation:**

Please kindly see above

---

### Decision · Program_Chairs · 2023-04-09

**Decision:**

No revision received; not invited to archive

**Comment:**

Please include your URM statement.